# Non-anaesthetist-administered ketamine for emergency caesarean section in Kenya: cost-effectiveness analysis

Stephen Charles Resch [1], Sebastian Suarez,[2,3]
Moshood Olanrewaju Omotayo,[2,4] Jennifer Griffin,[5] Daniel Sessler [6],
Thomas Burke[2,4,7]

For numbered affiliations see end of article.

**Correspondence to**
Dr Stephen Charles Resch;
sresch@hsph.harvard.edu

## ABSTRACT

**Objectives** Lack of anaesthesia services is a frequent barrier to emergency surgeries such as caesarean delivery in Kenya. This study aimed to estimate the survival gains and cost-effectiveness of scaling up the Every Second Matters (ESM)-Ketamine programme that trains non-anaesthetist providers to administer and monitor ketamine during emergency caesarean deliveries.

**Setting** Hospitals in Kenyan counties with low rates of caesarean delivery.

**Participants** Patients needing emergency caesarean delivery in settings without availability of standard anaesthesia service.

**Interventions** Simulated scales up of the ESM-Ketamine programme over 5 years (2020–24) was compared with status quo.

**Outcome measures** Cost of implementing the programme and corresponding additional emergency caesarean deliveries. Maternal and fetal/neonatal deaths prevented, and corresponding life-years gained due to increased provision of emergency caesarean procedures. Cost-effectiveness was assessed by comparing the cost per life-year gained of the ESM-Ketamine programme compared with status quo.

**Results** Over 5 years, the expected gap in emergency caesarean deliveries was 157 000. A US$1.2 million ESM-Ketamine programme reduced this gap by 28 700, averting by 316 maternal and 4736 fetal deaths and generating 331 000 total life-years gained. Cost-effectiveness of scaling up the ESM-Ketamine programme was US$44 per life-year gained in the base case and US$251 in the most pessimistic scenario—a very good value for Kenya at less than 20% of per capita GDP per life-year gained.

**Conclusion** In areas of Kenya with significant underprovision of emergency caesarean delivery due to a lack of availability of traditional anaesthesia, an ESM-Ketamine programme is likely to enable a substantial number of life-saving surgeries at reasonable cost.

## INTRODUCTION

The World Health Organization (WHO) estimates that at least 10%–15% and possibly as many as 19%[1,2] of all childbirths require caesarean deliveries, most on an emergency basis. Few low-income/middle-income countries provide this level of caesarean

## STRENGTHS AND LIMITATIONS OF THIS STUDY

⇒ This study uses a decision analytical approach that allows synthesis of data on costs and outcomes from a long-running Every Second Matters (ESM)-Ketamine programme implemented in one county in Kenya and use a simulation model to extrapolate the impact and cost-effectiveness of expanding the programme throughout the country.

⇒ Another strength of this method is that we are able to translate the observed emergency caesarean deliveries enabled by the ESM-Ketamine programme into maternal and fetal deaths averted and life-year gains using the MANDATE model to simulate outcomes of deliveries with complications when surgery is available and for a counterfactual scenario where surgery is not available.

⇒ A limitation of this study is the uncertainty about the extent to which lack of available anaesthesia is the key barrier to timely provision of emergency surgeries in Kenyan health facilities.

⇒ The study is also limited by uncertainty regarding the pace at which the supply of conventional anaesthesiology services can be expanded, such that the ESM-Ketamine programme would no longer be needed.

⇒ The study was limited to considering impact on childbirth-related mortality and did not capture potential additional benefits related to reductions in morbidity or mortality from other causes requiring emergency surgery.

deliveries, with considerable consequent morbidity and mortality. Kenya provides more caesarean deliveries than most sub-Saharan African countries, with a population rate of 8.7%.[3,4] However, this average rate masks substantial heterogeneity. In 16 of 47 counties the population caesarean delivery rate is below 5%. In 26 counties the rate is between 5% and 15%. In just five counties, mostly those with large urban areas such as Nairobi, the population caesarean delivery rate is over 15%. While rigorous data are

lacking, a substantial portion of caesarean procedures contributing to the rate observed in urban areas may be elective.[5]

Undersupply of emergency caesarean deliveries in many parts of Kenya likely contributes to the country's high maternal and perinatal mortality. When last measured in 2014, the maternal mortality ratio in Kenya was 362 (95% CI 254 to 471) per 100 000 births.[4] Perinatal mortality, defined as pregnancy losses occurring after seven completed months of gestation (stillbirths) plus deaths to live births within the first 7 days of life (early neonatal deaths), was 29 per 1000 births in 2014.[4 6]

One of the major limitations to provision of caesarean deliveries is lack of anaesthesia. The Lancet Commission on Global Surgery indicated substantially improved health outcomes associated with increases in specialist surgical workforce (surgeons, anaesthetists and obstetricians) up to at least 20 per 100 000 population, and the World Federation of Societies of Anaesthesiologists has indicated about half of these providers—that is, 10 per 100 000 population—should be anaesthesia providers.[7 8] In Kenya, there are less than two anaesthesia providers per 100 000 population, of which about 80% are non-physician anaesthetists including clinical officers and nurses.[7] The lack of anaesthesia services for emergency surgery is especially grave when considering that the workforce is maldistributed within the country in proportion to the population.[9]

We have previously described the Every Second Matters for Emergency and Essential Surgery-Ketamine (ESM-Ketamine) programme, which has been operating in rural Kenya since 2013.[10 11] While not envisioned as a long-term solution to the shortage of anaesthesia services in Kenya, or a substitute for other health system strengthening efforts aimed at increasing the supply and improving the distribution of anaesthetists and anaesthesiologists, the ESM-Ketamine programme was designed as a rapid, low-cost approach to ensuring anaesthesia is available for surgery in emergency situations. Through this programme, non-anaesthetist providers are trained to administer and monitor ketamine during emergency surgeries such as caesarean delivery when no anaesthetist is available, enabling procedures that would have not been possible, would have been significantly delayed, or would have been performed without any anaesthesia. The programme includes an intensive hands-on 5-day training, ESM-Ketamine kits, wallcharts, checklists and regular supervisory visits for quality assurance (QA). The ESM-Ketamine pilot programme in Kenya has proven remarkably safe, with positive patient experiences, no deaths or major adverse events attributed to ketamine in more than 2000 emergency and essential surgeries including about 450 caesarean deliveries across 17 hospitals.[12 13] However, there are distinct training, implementation and running costs associated with the ESM-Ketamine programme. In this analysis, we model the potential health impact and cost-effectiveness of scaling up the ESM-Ketamine programme for emergency caesarean sections throughout Kenya as compared with the status quo.

## METHODS

Our decision analytical approach, shown in figure 1, synthesised data from numerous sources to simulate the health impact and cost of scaling up the ESM-Ketamine programme nationally in Kenya. Using data from the 2014 Kenya Demographic and Health Survey,[4] we estimated the facility caesarean delivery rate (caesarean delivery performed divided by deliveries with a skilled provider) for all counties in Kenya and identified those where the facility caesarean delivery rate was less than 15%. For these counties, we estimated the number of facility-based deliveries over a 5-year period (2020–2024) using UN projections of annual births and calculated the county-specific gap in emergency caesarean delivery using an expected unconstrained emergency caesarean delivery rate of 15% as the reference level.

Ketamine-based anaesthesia via providers trained in the ESM-Ketamine programme addresses only one of several possible bottlenecks to emergency caesarean delivery. Thus, the impact of the programme will depend on the frequency with which the availability of traditional anaesthesia is the sole bottleneck. While no studies have measured this precisely at a patient case level, there is evidence that in some settings in Kenya, anaesthesia service is in shorter supply than other necessary service components for emergency caesarean. A 2011 study in Nyanza region found that for 34 operating theatres with associated doctors able to perform caesarean delivery, there were only 18 anaesthetists and 44% of theatres reported severely limited access to anaesthetist services.[14] More recently, in 2014, we surveyed all 30 operating theatres in 2 counties of Western Kenya and found 57% had no access to anaesthesia services.[15] Based on this evidence, we assumed that anaesthesia availability was the sole bottleneck in 30% of the cases in which an emergency caesarean delivery could not be provided.

We modelled a gradual 5-year scale up of the ESM-Ketamine programme that reaches full coverage of all facilities with operating rooms in the target counties (table 1). Based on the level of scale to be achieved each year, the number of additional emergency caesarean deliveries was calculated for each of the 5 years as compared with a status quo with no change in caesarean availability.

We used the previously developed, publicly available MANDATE model (www.mandate4mnh.org) to estimate the expected number of lives saved per additional emergency caesarean delivery in Kenya. This model synthesises evidence on the incidence, case fatality rate and efficacy of caesarean delivery for all major delivery complications for which emergency caesarean delivery is the recommended intervention.[16 17] We simulated two scenarios with this model: a base case approximating status quo for Kenya in 2017[10] and an alternative case where caesarean delivery is available (and used when indicated) in 99% of

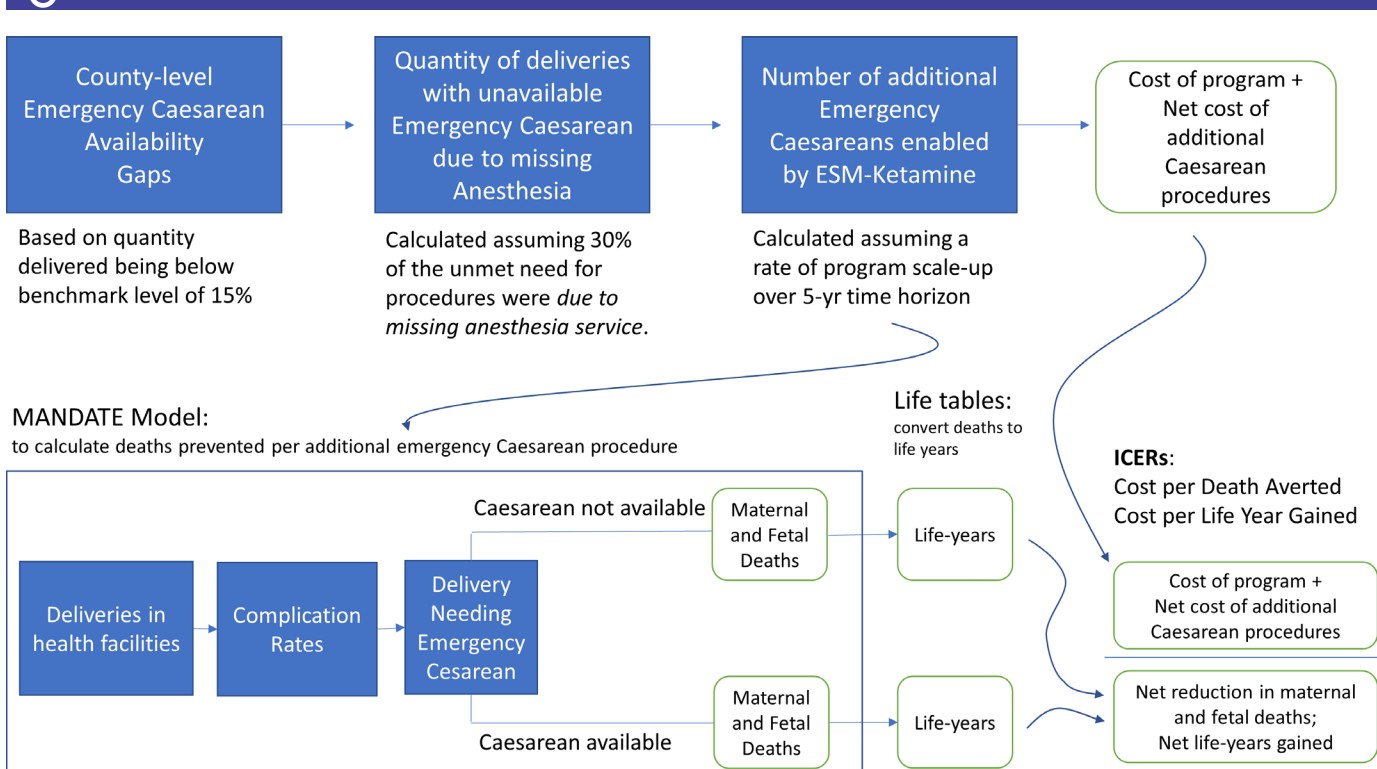

**Figure 1** Schematic diagram of decision analysis model. ESM, every second matters; ICER, incremental cost-effectiveness ratio.

emergency cases in hospital settings. See online supplemental table S1 for a detailed list of the baseline penetration and utilisation rates of caesarean section for each condition, as well as the corresponding efficacy parameters. We calculated the difference in caesarean delivery procedures performed and the difference in fetal and maternal deaths, and then calculated a ratio of deaths prevented per additional caesarean delivery procedure. We applied these ratios to the number of additional caesarean deliveries we estimated would occur in Kenya each year because of the scale-up of the ESM-Ketamine programme.

We converted deaths averted to years of life gained using the most recently available 2016 Kenya life tables. For fetal deaths prevented, we used the average of male and female life expectancy at less than 1 year of age. For maternal deaths prevented, we used the average of remaining life expectancy for three female age groups (20–24, 25–29, 30–34 years).

Evaluation of the ESM-Ketamine programme in 2059 Kenyan patients identified only occasional mild adverse events associated with ketamine such as hallucinations and salivation, and no major adverse events.[12] We; therefore, assumed that ESM-Ketamine does not cause major morbidities and did not include the minor consequences of the occasional mild and time-limited adverse events since they pale in comparison to the consequences of not providing emergency caesarean delivery.

## Costs

Based on the experience of the current ESM-Ketamine programme in Kenya, we modelled the health-sector cost of scaling up access. We assumed three ESM-Ketamine trained providers would be required in each facility with surgical capacity to ensure that availability of anaesthesia services would not be a bottleneck to emergency caesarean delivery. In the first year, 72 providers are expected to graduate from the training programme. In subsequent years, a second training centre is added, and the number of providers trained per year increases to 144. In the fifth year, the number of newly trained reduces as the necessary supply of ESM-Ketamine providers is reached. We assumed a 7% annual staff turnover rate and included additional training to maintain target supply of active ESM-Ketamine providers. The unit cost of training one provider was US$650 plus US$150 for travel and lodging. In addition, each facility requires a US$150 ESM-Ketamine kit and table 1 and online supplemental table S2. We included cost for supervision and coordination, QA activities and overhead. These costs included a full-time programme director, one QA officer per 60 facilities and 15% overhead.

The cost of additional caesarean delivery procedures (US$367 per delivery) was based on a recent rigorous cost analysis conducted in Rwanda adjusted for Kenya based on differences in purchasing power parity.[18] Because some of cost would be incurred even if no caesarean procedure was performed, we subtracted the cost of a vaginal delivery—which we assumed was half the cost of a caesarean delivery.[19]

**Table 1** Key model inputs

| Parameter | Base case value | Alternative values | Source notes |
|---|---|---|---|
| Demographics | | | |
| Estimated deliveries (2020) | 1.503 million | | Derived from crude birth rate and total population estimatein UN Population Prospects[25] |
| Change in total deliveries | 0.01 | | Derived from births in UN Population Prospects[25] |
| Target emergency caesarean delivery rate | 15% | 10% | |
| Share of emergency caesarean delivery gap attributable only to lack of anaesthesia services | 30% | 10% | 14, 15 |
| Training Programme Implementation Assumptions | | | |
| Training centres | One in first year, 2 thereafter | | |
| Providers trained per session | 6 | | |
| Trainings per year per training centre | 12 | | |
| ESMK provider turnover rate | 0.07 | | |
| ESMK providers per facility | 3 | | |
| ESM-ketamine programme unit costs | | | |
| Training | | | |
| Training costs | $650 | | TTS grant |
| Travel and lodging | $150 | | MGH/USAID |
| Kit (one per facility) | $150 | | TTS grant |
| Programme Coordination | | | |
| Programme director | $48 000 | | TTS grant |
| QA/QC | | | |
| Facilities per staff person | 60 | | MGH/USAID |
| QA/QC staff salary | $16 400 | | TTS grant |
| Overhead | 15% | | TTS grant |
| Life expectancy from age x | Male | Female | WHO Life Tables Kenya 2015[26] |
| <1 year | 64.4 | 68.9 | |
| 20–24 years | | 53.3 | |
| 25–29 years | | 48.8 | |
| 30–34 years | | 44.4 | |
| Life expectancy per averted death | Undiscounted | Discounted | |
| Mothers (average of 3 age groups) | 48.8 | 26.2 | |
| Newborns (average of male and female <1 year) | 66.7 | 29.5 | |
| Discount rate, annual | 3% | | |

ESMK, Every Second Matters-Ketamine; QA/QC, quality assurance/quality control; TTS, Saving Lives at Birth Partners: Transition to Scale.

## Cost-effectiveness analysis

To calculate the incremental cost per life-year gained, we divided the 5-year programme cost (in 2017 US dollars) by the number of life-years gained due to lives saved in the 5-year period, discounting both costs and health outcomes by 3% per year. We also reported undiscounted results. Kenya's GDP per capita in 2018 was US$1620.[20] Considerable debate persists about value thresholds for health interventions in settings like Kenya. The WHO has historically argued that interventions with an incremental cost-effectiveness ratio (ICER) less than per-capita gross domestic product (GDP) per disability-adjusted life-year (DALY) averted are cost-effective. More recently, researchers have advocated that much more stringent health-opportunity-loss based thresholds, are more appropriate when deciding to invest in new health technologies where health budgets are fixed.[21] Under this approach, interventions costing less than 25% of GDP per capita per DALY averted are generally considered highly cost-effective by most standards.[22] Our model estimated life-years gained but could not account for disability in those years. Therefore, we used 20% GDP per capita

per life-year gained (US$325 in Kenya) as a benchmark for good value, which is approximately equivalent to assuming an average lifetime health-related quality of life of 0.85 (where one represents perfect health and zero represents death).[23]

### Patient and public involvement

Our study uses aggregated secondary data from the ESM-Ketamine programme in Kenya regarding outcomes for patients undergoing surgery with ketamine anaesthesia. But there is no patient involvement in this model-based cost-effectiveness study.

### Data availability

No additional data available.

### Sensitivity analysis

The impact and value of the ESM-Ketamine programme will depend on the actual number of emergency caesarean deliveries that are enabled by the programme, which in turn depends both on the number of emergency cases requiring caesarean delivery that are currently not receiving caesarean delivery as well as the portion of these cases in which lack of anaesthesia is the sole bottleneck preventing the caesarean delivery procedure from being performed. In sensitivity analysis, we considered scenarios in which only 10% of all hospital deliveries require emergency caesarean delivery (vs 15% rate assumed in base case), and scenarios in which lack of anaesthesia was the sole bottleneck for only 10% of cases in which emergency caesarean delivery was indicated but not provided (vs 30% in the base case). We also considered the case in which the ESM-Ketamine programme costs and the additional delivery costs associated with caesarean section were twice as high as in the base case.

## RESULTS

The results are reported according to the specifications of the Consolidated Health Economic Evaluation Reporting Standards statement as documented in the checklist (online supplemental table S2).

### Program scope and cost

In Kenya, 33 counties met the criteria of having a facility caesarean delivery rate less than 15% (online supplemental table S3). The emergency caesarean delivery gap across these counties was 157 000 procedures over 5 years, of which 47 000 are attributable to lack of anaesthesia. The target counties contain 191 hospitals with operating rooms. The modelled ESM-Ketamine programme would train 693 providers, enough to maintain 3 providers per facility, while accounting for staff turnover (online supplemental table S4). The scale up of the ESM-Ketamine programme from 13% coverage in year 1 to 100% coverage in year 5 would enable 28 700 emergency caesarean deliveries, reducing the overall gap in emergency caesarean delivery by 18% and the anaesthesia-attributable gap by 61%. The undiscounted cost of the 5-year scale up of the ESM-Ketamine programme is estimated to be about US$1.2 million (online supplemental table S5) or US$1700 per ESM-Ketamine provider trained. About half of this cost is the direct cost of training providers (online supplemental figure S1). The cost of the ESM-Ketamine programme would be and US$41 per emergency caesarean delivery enabled. The total incremental cost, when including the additional cost of the caesarean procedure itself, is about US$224 per delivery.

### Health outcomes

Results from the MANDATE model indicate that the most common indications for emergency caesarean delivery are obstructed labour, pre-eclampsia or eclampsia, and fetal distress (table 2). In the baseline scenario, the hospital caesarean delivery rate was 7.1%. By maximising the availability and use of caesarean delivery for emergency indications in hospital settings (without changing care-seeking patterns or the effectiveness of the emergency medical transport system) the number of caesarean deliveries almost doubles—to a rate of 12.3%—which corresponds well to the level of emergency caesarean delivery expected based on the incidence of life-saving indications found in an observational study.[24]

Analysing the corresponding reduction in maternal and fetal death for improved access to caesarean delivery compared with the status quo, we found that 0.178 deaths would be averted per additional emergency caesarean procedure provided and (table 2 and online supplemental table S6). Over 90% of these deaths were fetal. One fetal death was prevented by every 6.1 emergency caesarean delivery procedures. One maternal death was averted by every 92 emergency caesarean delivery procedures. These estimates represent a weighted average across the emergency indications and accounts for condition-specific mortality risk reduction conferred by caesarean delivery (online supplemental table S7).

Applying these rates of life-saving to the additional 28 700 caesarean deliveries provided over 5 years as a result of the scale-up of the ESM-Ketamine programme, we estimated that maternal and fetal deaths would be reduced by 316 and 4736, respectively (table 3). Considering remaining life expectancies at the moment these deaths are averted, the mortality reductions would translate into 331 000 total life-years gained.

### Cost-effectiveness

Cost-effectiveness results are shown in (table 4). Without discounting, the average cost per death prevented by ketamine-enabled caesarean delivery performed during the 5-year period in the base case was about US$1270. Considering the expected years of life gained by preventing deaths, and discounting both costs and future life-years, the cost-effectiveness of scaling up the ESM-Ketamine programme was US$44 per life-year gained.

In sensitivity analysis, when the caesarean delivery gap among hospital deliveries is based on a bottleneck-free caesarean delivery rate of 10%, and the portion of the gap

Table 2 Number of emergency caesarean delivery procedures by indication under a baseline scenario and an 'Improved' scenario in which there are no bottlenecks to emergency caesarean delivery in hospital facilities*

| Indication | Baseline | Improved | Difference | % of total difference |
|---|---|---|---|---|
| Obstructed labour | 13214 | 18330 | 5116 | 36 |
| Pre-eclampsia/eclampsia | 2241 | 5469 | 3228 | 23 |
| AIPH-abruption | 495 | 1280 | 785 | 6 |
| AIPH-placenta previa | 166 | 422 | 256 | 2 |
| AIPH-ruptured uterus | 231 | 432 | 201 | 1 |
| SFD-MB | 73 | 242 | 169 | 1 |
| SFD-IUGR | 698 | 2327 | 1629 | 11 |
| SFD-breech | 73 | 241 | 168 | 1 |
| SFD-cord | 284 | 949 | 665 | 5 |
| SFD-other | 837 | 2789 | 1952 | 14 |
| Total procedures | 18312 | 32481 | 14169 | 100 |
| Deaths Averted | | | | |
| Maternal | | | 154 | 6.2 |
| Fetal | | | 2339 | 93.8 |
| Total | | | 2493 | |
| Deaths averted per emergency caesarean | | | | |
| Maternal | | | 0.011 | |
| Fetal | | | 0.165 | |
| Total | | | 0.178 | |

*The total number of hospital deliveries was 258 630. In the 'baseline' scenario, the emergency caesarean delivery rate is about 7.1% for hospital deliveries, and in the 'improved scenario'—representing 99% coverage of emergency caesarean delivery in hospital setting—the emergency caesarean delivery rate increases to 12.3% of hospital deliveries. In both scenarios, the analysis assumed no non-emergency caesarean deliveries occur for reasons such as maternal request, revenue maximisation or scheduling convenience.
AIPH, ante/intrapartum haemorrhage; IUGR, intrauterine growth rate; MB, multiple births; SFD, significant fetal distress.

attributable to lack of anaesthesia is only 10%, an ESM-Ketamine programme would prevent 28 maternal and 426 fetal deaths, translating to 29 750 life-years gained, and a cost-effectiveness of US$125 per life-year gained. In the most pessimistic scenario considered, combining lower impact on caesarean section uptake, higher caesarean procedure cost, and a doubling of the cost of the ESM-Ketamine programme itself, the cost-effectiveness ratio would increase to US$251 per life-year gained.

## DISCUSSION

Our analysis shows an ESM-Ketamine programme could enable an additional 28 700 emergency caesarean deliveries over 5 years, reducing the projected gap in emergency caesarean deliveries in 33 Kenyan counties by about 18%. The cost of the ESM-Ketamine programme would be about US$1700 per ESM-Ketamine provider trained and US$41 per emergency caesarean delivery enabled. The total incremental cost, when including the additional cost of the caesarean procedure itself, is about US$224 per delivery.

Given that we found one death is expected to be prevented by every six emergency caesarean delivery procedures, the programme is likely to be highly cost-effective. In the base case, the scale up of ESM-Ketamine programme over 5 years had a net cost of US$5.9 million and resulted in 5052 deaths averted, translating to an ICER of US$44 per life-year gained.

There is substantial uncertainty in the data that underlies our analysis, which could limit confidence in the results. The impact of the ESM-Ketamine programme on clinical outcomes, compared with a status quo counterfactual, has not been measured in a randomised controlled trial. Therefore, we modelled the number of emergency caesarean procedures enabled by ESM-Ketamine, as well as the clinical outcomes for deliveries requiring caesarean when ketamine is available and when it is not. To address the uncertainty in our model, we tested the sensitivity of our conclusions about the value of the ESM-Ketamine programme over a wide range of less favourable assumptions. In the most pessimistic scenario we considered—in which the number of caesarean deliveries enabled was reduced to 2580 (9% of the base case amount) and the cost of both the ESM-programme itself and the cost of caesarean deliveries was doubled, the cost per life-year gained only increased to US$251. In all scenarios, the cost-effectiveness ratios were far less than 20% of GDP per capita ($325) benchmark we used as a threshold value and

**Table 3** Programmatic and health outcomes by year

| Year | 2020 | 2021 | 2022 | 2023 | 2024 | Total |
|---|---|---|---|---|---|---|
| Emergency CS gap attributable to anaesthesia | 9248 | 9341 | 9434 | 9528 | 9624 | 47175 |
| Coverage scale-up | 13% | 38% | 63% | 88% | 100% | 61% |
| Additional CS provided | 1202 | 3549 | 5943 | 8385 | 9624 | 28704 |
| Lives saved | | | | | | |
| Maternal | 13 | 39 | 65 | 92 | 106 | 316 |
| Fetal | 198 | 586 | 981 | 1384 | 1588 | 4736 |
| Total | 212 | 625 | 1046 | 1476 | 1694 | 5052 |
| Life-years gained | | | | | | |
| Maternal | 646 | 1907 | 3193 | 4504 | 5170 | 15419 |
| Newborn | 13222 | 39034 | 65362 | 92212 | 105834 | 315664 |
| Total | 13867 | 40941 | 68554 | 96716 | 111004 | 331083 |
| Economic outcomes | | | | | | |
| ESMK programme | US$149328 | US$248343 | US$258118 | US$286753 | US$237188 | US$1179730 |
| Additional CS | US$219791 | US$648892 | US$1086553 | US$1532902 | US$1759353 | US$5247491 |
| CS share of total cost | 60% | 72% | 81% | 84% | 88% | 82% |
| Total cost | US$369119 | US$897235 | US$1344671 | US$1819655 | US$1996541 | US$6427221 |

Base case, undiscounted.
CS, emergency caesarean; ESMK, Every Second Matters- Ketamine.

compare favourably to many public health interventions being implemented in Kenya. Indeed, using US$325 per life-year as a threshold for good value, the ESM-Ketamine programme would only have to enable 1720 emergency caesarean deliveries and avert about 300 deaths in 5 years to be considered cost-effective—which is only about 6% of the impact we estimate.

Our analysis assumed that there would be no change in the portion of deliveries in health facilities, no change in the availability of timely emergency transport to hospitals with capacity for caesarean delivery, and no reduction in the caesarean delivery gap due to increasing availability of traditional anaesthesia services. If more deliveries were to occur in health facilities over time, then there would be more opportunities for the ESM-Ketamine programme to enable life-saving emergency caesarean delivery. If the expansion of traditional anaesthesia service reduces the gap in caesarean delivery, the ESM-Ketamine programme would be less cost-effective and, ideally, unnecessary. However, the payback period for an ESM-Ketamine programme is very short. It is highly unlikely that the investment in establishing a programme and training ESM-Ketamine providers would be rendered moot by a sudden expansion of traditional anaesthesia services. For example, in the base case, each ESM-Ketamine trained provider would enable about 41 emergency caesarean deliveries on average over the 5-year time horizon, resulting in about 7.3 lives saved per ESM-Ketamine provider trained. Yet, to meet standard benchmarks for cost-effectiveness, each ESM-Ketamine provider would only need to enable about three emergency caesarean

deliveries—a number that might reasonably be expected within a few months of completing training.

The expected cost of the ESM-Ketamine programme is small relative to the cost of caesarean delivery procedures enabled by the availability of ketamine anaesthesia. Therefore, the value of the ESM-Ketamine programme is driven in large part by the value of emergency caesarean delivery itself. The overall cost-effectiveness of ESM-Ketamine was much more sensitive to the additional cost of caesarean delivery compared with vaginal delivery than the cost of the ESM-Ketamine programme itself.

Another limitation in our analysis is the scope of benefits considered. Although caesarean delivery is also likely to prevent morbidity in both mothers and newborns (eg, long-term cognitive problems associated with non-fatal birth asphyxia), we did not include these health benefits. We also did not consider the value of the ESM-Ketamine programme related to surgical procedures other than emergency caesarean deliveries. In the ESM-Ketamine programme in Kenya, emergency caesarean deliveries account for about 20% of the total procedures performed with many of the remaining procedures being for acute abdomens, open fractures and similarly serious conditions. For these reasons, it is likely that the total value of the ESM-Ketamine programme is larger than we estimated.

Our analysis supports the relatively modest investment needed for scaling up the ESM-Ketamine programme over the next 5 years. Cost per life-year gained ranged from US$44 (2.8% of GDP per capita) in the base case to US$251 (16% of GDP per capita) in our most pessimistic

**Table 4** Cost-effectiveness results with sensitivity analysis for key model uncertainties

| | Base case impact | Base case+higher CS cost | Base case+higher CS cost and ESMK programme cost | Pessimistic impact | Pessimistic impact+higher CS cost | Pessimistic impact+higher CS cost and ESMK programme cost |
|---|---|---|---|---|---|---|
| **Input parameters** | | | | | | |
| Population rate of emergency CS | 15% | 15% | 15% | 10% | 10% | 10% |
| Emergency CS gap attributable to anaesthesia | 30% | 30% | 30% | 10% | 10% | 10% |
| Incremental CS procedure cost* | US$183 | US$366 | US$366 | US$183 | US$366 | US$366 |
| Programme cost multiplier | 1× | 1× | 2× | 1× | 1× | 2× |
| **Outcomes** | | | | | | |
| Undiscounted programme cost | US$1.18m | US$1.18m | US$2.36m | US$1.18m | US$1.18m | US$2.36m |
| Undiscounted total cost | US$6.43m | US$11.68m | US$12.85m | US$1.65m | US$2.12m | US$3.30m |
| Additional emergency CS | 28704 | 28704 | 28704 | 2580 | 2580 | 2580 |
| Maternal lives saved | 316 | 316 | 316 | 28 | 28 | 28 |
| Fetal lives saved | 4736 | 4736 | 4736 | 426 | 426 | 426 |
| **Incremental cost-effectiveness ratio** | | | | | | |
| Cost per LY (discounted) | US$44 | US$79 | US$87 | US$125 | US$161 | US$251 |
| Percent of GDPpc per LY (discounted) | 2.8% | 5.1% | 5.6% | 8.0% | 10.3% | 16.1% |
| Cost per maternal LY (discounted) | US$779 | US$1412 | US$1557 | US$2246 | US$2880 | US$4492 |
| Percent of GDPpc per maternal LY (discounted) | 50% | 91% | 100% | 144% | 185% | 288% |

*Excess cost of caesarean delivery cost above the cost of normal vaginal delivery.
CS, caesarean delivery procedure; ESMK, Every Second Matters–Ketamine; GDPpc, per capita gross domestic product; LY, life-year.

scenario. There remains significant uncertainty about the overall scale of the impact of an ESM-Ketamine programme due to a lack of data regarding the extent to which anaesthesia is a key bottleneck preventing access to emergency caesarean delivery. Nevertheless, we found that the ESM-Ketamine programme would be worthwhile even if the number of emergency caesarean procedures it enabled were an order-of-magnitude smaller than expected in our base case. In areas of Kenya with significant underprovision of emergency caesarean delivery due to a lack of availability of traditional anaesthesia, an ESM-Ketamine programme is likely to enable a substantial number of life-saving surgeries at modest cost.

**Author affiliations**
[1]Center for Health Decision Science, Harvard University T H Chan School of Public Health, Boston, Massachusetts, USA
[2]Division of Global Health and Human Rights, Department of Emergency Medicine, Massachusetts General Hospital, Boston, Massachusetts, USA
[3]Boston University, Boston, Massachusetts, USA
[4]Harvard Medical School, Boston, Massachusetts, USA
[5]Center for Global Health, RTI International, Research Triangle Park, North Carolina, USA
[6]Department of Outcomes Research, Cleveland Clinic, Cleveland, Ohio, USA
[7]Harvard University T H Chan School of Public Health, Boston, Massachusetts, USA

**Contributors** The study was conceptualised and designed by SCR, TB and SS. SS and MOO assisted in gathering information from the ESM-ketamine pilot project to inform parameters of the cost-effectiveness model. JG conducted analysis with the Mandate model to inform parameters regarding the impact of emergency caesarean section on neonatal and maternal deaths. SCR developed the cost-effectiveness model and conducted the analysis. SCR, with DS and TB, wrote the manuscript. All authors reviewed and edited manuscript drafts. SCR and TFB are the guarantors of the manuscript and accept full responsibility for the work and conduct of the study, had access to the data, and controlled the decision to publish. The corresponding author attests that all listed authors meet authorship criteria and that no others meeting the criteria have been omitted.

**Funding** The work presented in this manuscript was funded by USAID Award No. 7200AA18CA00002 (project title: 'Scaling Up Every Second Matters for Mothers and Babies-Ketamine (ESM-Ketamine) in Kenya') and supported by the Ujenzi Charitable Trust (Grant number: Not Applicable), Elrha's Research for Health in Humanitarian Crises (R2HC) Program (Grant number: Not Applicable) and the Saving Lives at Birth partners (Grant number: Not Applicable). The Saving Lives at Birth partners are: United States Agency for International Development (USAID), the Government of Norway, the Bill & Melinda Gates Foundation, Grand Challenges Canada, the UK Government, and the Korea International Cooperation Agency (KOICA). The R2HC program is funded by the UK Foreign, Commonwealth and Development Office (FCDO), Wellcome, and the UK National Institute for Health Research (NIHR). This manuscript was prepared by the listed authors and does not necessarily reflect the views of the funding partners.

**Disclaimer** The funding sources had no role in the collection, analysis, or interpretation of the data nor in manuscript preparation.

**Competing interests** None declared.

**Patient and public involvement** Patients and/or the public were not involved in the design, or conduct, or reporting, or dissemination plans of this research.

**Patient consent for publication** Not applicable.

**Ethics approval** This model-based study did not collect any new information from human subjects or use any patient-level data, so we did not obtain approval of an Institutional Review Board.

**Provenance and peer review** Not commissioned; externally peer reviewed.

**Data availability statement** Data are available on reasonable request. All the data used in this analysis are publicly available. An excel workbook of data is available on request.

**ORCID iDs**
Stephen Charles Resch http://orcid.org/0000-0002-0858-5467
Daniel Sessler http://orcid.org/0000-0001-9932-3077

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
