## [Reviewer comments · BMJ Open]

ARTICLE DETAILS

TITLE (PROVISIONAL)	Non-anaesthetist Administered Ketamine for Emergency Caesarean Section in Kenya: Cost-effectiveness Analysis
AUTHORS	Resch, Stephen; Suarez, Sebastian; Omotayo, Moshood Olanrewaju; Griffin, Jennifer; Sessler, Daniel; Burke, Thomas

VERSION 1 – REVIEW

REVIEWER	Rasooli, Fatemeh Jondishapour University of Medical Sciences, Emergency Medicine
REVIEW RETURNED	22-Feb-2022

GENERAL COMMENTS	Thank you for your valuable article. I have some questions and comment about your study: 1) In the abstract, it is not clear if this is a simulated study and this continues to the end of this part.2) In the “objective” part, it is stated that” This study aimed to estimate the health impact and cost-effectiveness of scaling up the Every Second Matters (ESM)...” , we know that the health impact of ESM program on emergent deliveries in Kenia has been evaluated in many former studies so it would be better that this study focuses more on costs.3) In “outcome measure” part: it is mentioned that “Cost-effectiveness was assessed by comparing the cost per life-year gained of the ESM-Ketamine program compared to status quo.”, Is there no actual data on the cost of ESM program in Kenya regarding the amount of performed studies until now?4) page 5, paragraph 2, line 6 is stated“ the ESM-Ketamine program is a relatively inexpensive and rapid approach to ensure anaesthesia is available for surgery in emergency situations“ , is there any reference for this (costs) ?4) “ method” part: there are a lot of information in this part that can be just referenced and concentrate deeper on method.5) Ethical approval: there is no overall consensus on ethical approval in such studies, so the final decision depends on the policy of the journal.5) the type of study and related checklist not mentioned in the explanations. Note: there wasn’t reference part in the submitted article for me, so I could not able to assess this part.
---

REVIEWER	Kuznik, Andreas Regeneron Pharmaceuticals Inc, Health Economics and Outcomes Research
REVIEW RETURNED	23-Feb-2022

GENERAL COMMENTS	Peer Review: Cost-effectiveness of Ketamine Anaesthesia for
---

Emergency Caesarean Section in Kenya

It seems plausible that the availability of a safe and effective anesthetic for emergency C-Section at childbirth in cases where it is medically needed would be vastly superior to the alternative. Assuming that maternal and neonatal health could be improved by means of this intervention it seems furthermore plausible that ketamine anaesthesia would be highly cost-effective, since it is associated with a relatively low, one-time cost, whereas the benefits of a healthy newborn or a healthy mother would continue to accrue over many years. In that sense, the authors' findings have face validity.

Unfortunately, it is not at all transparent how the authors have arrived at their results and some of the findings even appear conflicting. I have listed several major and minor comments below that I feel would not to be addressed before this paper could be considered for publication.

Major comments:

1. The reference list is empty – this is a substantial inconvenience as I was unable to verify any of the claims that were made regarding the effectiveness of the ketamine anaesthesia program.
2. A 15% rate of medically necessary C-sections is used as the anchor in the model. It is not clear where this 15% estimate comes from. References 1 and 2 (which I was not able to verify) listed in the introduction provide a range of 10%-15%. Since the difference between 15% and the present rate of C-sections of 7.1% is used to scale up the results, this parameter is highly influential. This really should be informed by a robust reference that estimates the number of medically necessary C-sections in a sample of rural clinics through primary data collection. Apologies if this comment is redundant, but again, I am unable to pull up the list of references.
3. The calculations of how one medically necessary C-section translates into 0.178 deaths are not transparent enough. The Mandate model merely refers to a website where this number was presumably generated, but the math behind this calculation really belongs into the manuscript.
4. Why use life years in the denominator of the ICER if it is quite simple to use disability adjusted life years (DALYs)? The vast majority of cost-effectiveness analyses aimed at resource limited settings use DALYs in the denominator, which enables the reader to compare the value of interventions across therapeutic areas. It is not clear why the authors would not do the same.
5. The results do not add up. When you look at the national estimates, you have a total 5-year health system cost of \$1.2 million (page 12, line 37) and a benefit of 331,000 life years gained (page 13, line 44). That is \$3.62 per LY gained. However, in the cost-effectiveness section, the authors report a cost per life year gained of \$44 (page 14, line 7). How do I reconcile this difference?

Minor comments:

1. The authors define 20% of the Kenyan GDP to represent “good value”. This definition seems arbitrary. WHO-CHOICE defines an intervention to be highly cost-effective if the cost per DALY averted falls below one-time GDP. It is unclear where the 20% comes from.
2. Discussion, first paragraph, none of these numbers appear in the results section. It seems odd to discuss results that were not first presented in the results section.

VERSION 1 – AUTHOR RESPONSE

Reviewer: 1

Dr. Fatemeh Rasooli, Jondishapour University of Medical Sciences

Comments to the Author:

Dear author,

Thank you for your valuable article.

I have some questions and comment about your study:

1) In the abstract, it is not clear if this is a simulated study and this continues to the end of this part.

Response: In the abstract we added the word “Simulated” to clarify that this is a model-based simulation of what would happen over time in the future if the program was scaled up.

Interventions: Simulated scale up of the ESM-Ketamine program over five years (2020-24) was compared to status quo.

2) In the “objective” part, it is stated that “This study aimed to estimate the health impact and cost-effectiveness of scaling up the Every Second Matters (ESM)...”, we know that the health impact of ESM program on emergent deliveries in Kenya has been evaluated in many former studies so it would be better that this study focuses more on costs.

Response: While several studies have examined health impact measuring additional surgeries performed, our study adds information on health impact by estimating the deaths averted and life years gained by enabling additional emergency surgeries. To emphasize this feature, we replaced “health outcomes” with “survival gains”. As you note, we also focus on the value question of whether it is worthwhile considering the cost of the program.

This study aimed to estimate the survival gains and cost-effectiveness of scaling up the Every Second Matters (ESM) – Ketamine program

3) In “outcome measure” part: it is mentioned that “Cost-effectiveness was assessed by comparing the cost per life-year gained of the ESM-Ketamine program compared to status quo.”, Is there no actual data on the cost of ESM program in Kenya regarding the amount of performed studies until now?

Response: To our knowledge the cost of the ESM-Ketamine program has not been previously reported in peer reviewed literature.

4) page 5, paragraph 2, line 6 is stated “the ESM-Ketamine program is a relatively inexpensive and rapid approach to ensure anaesthesia is available for surgery in emergency situations”, is there any reference for this (costs)?

Response: The data on the cost of the program are reported in this manuscript and have not been described elsewhere. We deleted the phrase ‘relatively inexpensive’ here which implies a comparison, and rewrite the phrase to indicate that the ESM-Ketamine was designed to be rapid and low-cost.

...the ESM-Ketamine program was designed as a rapid, low-cost approach to ensuring anaesthesia is available for surgery in emergency situations.

4) “method” part: there are a lot of information in this part that can be just referenced and concentrate deeper on method.

Response: To address this issue, we deleted some words in order to make room for more methodological detail in the areas reviewers felt were important, in particular by adding a figure (Figure 1 separate file) on the overall modelling approach, as also recommended in CHEERS checklist, and providing greater explanation of the how we used the MANDATE model to estimate the deaths averted associated with an emergency Caesarean delivery.

5) Ethical approval: there is no overall consensus on ethical approval in such studies, so the final decision depends on the policy of the journal.

Response: Our study used only secondary data from publicly available sources, so it is not human subjects research.

5) the type of study and related checklist not mentioned in the explanations.

Note: there wasn't reference part in the submitted article for me, so I could not able to assess this part.

Response: We have added the CHEERS Checklist for this Economic Evaluation to the submitted materials.

Reviewer: 2

Dr. Andreas Kuznik, Regeneron Pharmaceuticals Inc

Comments to the Author:

Peer Review: Cost-effectiveness of Ketamine Anaesthesia for Emergency Caesarean Section in Kenya

It seems plausible that the availability of a safe and effective anesthetic for emergency C-Section at childbirth in cases where it is medically needed would be vastly superior to the alternative. Assuming that maternal and neonatal health could be improved by means of this intervention it seems furthermore plausible that ketamine anaesthesia would be highly cost-effective, since it is associated with a relatively low, one-time cost, whereas the benefits of a healthy newborn or a healthy mother would continue to accrue over many years. In that sense, the authors' findings have face validity.

Unfortunately, it is not at all transparent how the authors have arrived at their results and some of the findings even appear conflicting. I have listed several major and minor comments below that I feel would not to be addressed before this paper could be considered for publication.

Major comments:

1. The reference list is empty – this is a substantial inconvenience as I was unable to verify any of the claims that were made regarding the effectiveness of the ketamine anaesthesia program.

Response: We apologize for this oversight in document conversion. In the revision, we have corrected the omission of the bibliography reference list

2. A 15% rate of medically necessary C-sections is used as the anchor in the model. It is not clear where this 15% estimate comes from. References 1 and 2 (which I was not able to verify) listed in the introduction provide a range of 10%-15%. Since the difference between 15% and the present rate of C-sections of 7.1% is used to scale up the results, this parameter is highly influential. This really should be informed by a robust reference that estimates the number of medically necessary C-sections in a sample of rural clinics through primary data collection. Apologies if this comment is redundant, but again, I am unable to pull up the list of references.

Response: We used the best estimates of the rate of need for Caesarean section from the WHO Working Group on Caesarean Section. Recognizing, as the reviewer notes, that this input parameter is a significant driver of our model's results, and that there is some debate over the optimal rate of Caesarean section in the medical community, we conducted sensitivity analysis using values as low as 10% for the optimal rate of Caesarean section in Kenya.

3. The calculations of how one medically necessary C-section translates into 0.178 deaths are not transparent enough. The Mandate model merely refers to a website where this number was presumably generated, but the math behind this calculation really belongs into the manuscript.

Response: the calculations of the MANDATE model are very complex, but well documented on the public website, with additional detail available from the model developers. We have added more information to the supplemental material Table S6 explaining the key input parameters that we used in the MANDATE model analysis to estimate the number of deaths prevented per additional C section procedure performed when indicated for emergency delivery complications. We also added Figure 1 in the main text which shows how the MANDATE model fits into the overall analytic approach.

We added at Line 90:

See Table S6 for a detailed list of the baseline penetration and utilization rates of Caesarean section for each condition, as well as the corresponding efficacy parameters. We calculated the difference in caesarean delivery procedures performed (Table S6) and the difference in

foetal and maternal deaths (Table S5), and then calculated a ratio of deaths prevented per additional caesarean delivery procedure.

4. Why use life years in the denominator of the ICER if it is quite simple to use disability adjusted life years (DALYs)? The vast majority of cost-effectiveness analyses aimed at resource limited settings use DALYs in the denominator, which enables the reader to compare the value of interventions across therapeutic areas. It is not clear why the authors would not do the same.

Response: It is true that we would have preferred to use DALYs as our summary measure of health. This is the preferred outcome measure in CEAs in resource limited settings. However, we did not have data on the lifetime Years Lived with Disability (YLDs) a person in Kenya would experience if they did not die in childbirth. Moreover, the MANDATE model's scope only allowed us to estimate the number of deaths prevented by increased provision of Caesarean section. It does not estimate morbidity impacts. Finally, we believe the vast majority of the DALYs averted by preventing death in childbirth would be Years of Life Lost (YLLs) averted (i.e. survival gains). For these reasons, we used life-years instead of trying to calculate DALYs averted. Recognizing that the CEA value thresholds are based on DALYs, we did make a downward adjustment on the threshold used, since a life-year gained (if lived with some morbidity) is less valuable than a DALY averted.

5. The results do not add up. When you look at the national estimates, you have a total 5-year health system cost of \$1.2 million (page 12, line 37) and a benefit of 331,000 life years gained (page 13, line 44). That is \$3.62 per LY gained. However, in the cost-effectiveness section, the authors report a cost per life year gained of \$44 (page 14, line 7). How do I reconcile this difference?

Response: The \$1.2 million is the undiscounted 5-year program cost of the ESM-Ketamine program. To this must be added the undiscounted \$5.3 million extra delivery cost associated with additional Caesarean section deliveries over that 5-year period. (See Economic Outcomes panel of Table 3). When calculating the \$44 Cost per Life Year gained ICER shown in Table 4, we also discounted both costs and life years at 3% per year, as recommend in CEA guidelines.

Minor comments:

1. The authors define 20% of the Kenyan GDP to represent "good value". This definition seems arbitrary. WHO-CHOICE defines an intervention to be highly cost-effective if the cost per DALY averted falls below one-time GDP. It is unclear where the 20% comes from.

Response: We have added additional explanation regarding the choice of GDP-based cost-effectiveness threshold on page 10, line 132. In this, we acknowledge the WHO-CHOICE threshold of 1xGDP per capita per DALY averted, and explain why we use the more conservative health-opportunity loss approach, including a citation to a seminal paper on the topic. The source for the 25%*GDP per capita per DALY averted threshold we used is from the Beth Woods paper estimating health-opportunity-loss thresholds for low- and middle-income settings. Recently Ijeoma Edoaka led an empirical analysis that found a threshold of 0.5*GDP per capita per DALY averted for South Africa – a much wealthier country than Kenya. We think this further supports the use of thresholds that are a fraction of GDP per capita in these settings. In any case, the intervention we evaluated was highly cost-effective even with these very conservative thresholds.

page 10, line 132:

Considerable debate persists about value thresholds for health interventions in settings like Kenya. The WHO has historically argued that interventions with an ICER less than per-capita GDP per DALY averted are cost-effective. More recently, researchers have advocated that much more stringent health-opportunity-loss based thresholds, are more appropriate when deciding to invest in new health technologies where health budgets are fixed [21].

2. Discussion, first paragraph, none of these numbers appear in the results section. It seems odd to discuss results that were not first presented in the results section.

Response: The 28,700 enabled C-sections across 33 counties is reported in the Results section's first paragraph on page 12, as is 18% reduction in C Section gap. We added more explanation of the costs to the Results so that they are presented prior to the discussion.

Page 12, line 177-182:

The undiscounted cost of the five-year scale up of the ESM-Ketamine program is estimated to be about \$1.2 million (Table S3), or \$1700 per ESM-Ketamine provider trained. About half of this cost is the direct cost of training providers (Figure S1). The cost of the ESM-Ketamine program would be and \$41 per emergency caesarean delivery enabled. The total incremental cost, when including the additional cost of the caesarean procedure itself, is about \$224 per delivery.

VERSION 2 – REVIEW

REVIEWER	Kuznik, Andreas Regeneron Pharmaceuticals Inc, Health Economics and Outcomes Research
REVIEW RETURNED	20-Jun-2022

GENERAL COMMENTS	It seems plausible that the availability of a safe and effective anesthetic for emergency C-Section at childbirth in cases where it is medically needed would be vastly superior to the alternative. Assuming that maternal and neonatal health could be improved by means of this intervention it seems furthermore plausible that ketamine anaesthesia would be highly cost-effective, since it is associated with a relatively low, one-time cost, whereas the benefits of a healthy newborn or a healthy mother would continue to accrue over many years. In that sense, the authors' findings have face validity. Unfortunately, it is not at all transparent how the authors have arrived at their results and some of the findings even appear conflicting. I have listed several major and minor comments below that I feel would not to be addressed before this paper could be considered for publication. Major comments: 1. The reference list is empty – this is a substantial inconvenience as I was unable to verify any of the claims that were made regarding the effectiveness of the ketamine anaesthesia program. Response: We apologize for this oversight in document conversion. In the revision, we have corrected the omission of the bibliography reference list RESPONSE: Thanks 2. A 15% rate of medically necessary C-sections is used as the anchor in the model. It is not clear where this 15% estimate comes from. References 1 and 2 (which I was not able to verify) listed in the introduction provide a range of 10%-15%. Since the difference between 15% and the present rate of C-sections of 7.1% is used to scale up the results, this parameter is highly influential. This really should be informed by a robust reference that estimates the number of medically necessary C-sections in a sample of rural clinics through primary data collection. Apologies if this comment is redundant, but again, I am unable to pull up the list of references. Response: We used the best estimates of the rate of need for Caesarean section from the WHO Working Group on Caesarean Section. Recognizing, as the reviewer notes, that this input parameter is a significant driver of our model's results, and that there is some debate over the optimal rate of Caesarean section in the medical community, we conducted sensitivity analysis using values
---

as low as 10% for the optimal rate of Caesarean section in Kenya.
RESPONSE: That works, thanks.

3. The calculations of how one medically necessary C-section translates into 0.178 deaths are not transparent enough. The Mandate model merely refers to a website where this number was presumably generated, but the math behind this calculation really belongs into the manuscript.

Response: the calculations of the MANDATE model are very complex, but well documented on the public website, with additional detail available from the model developers. We have added more information to the supplemental material Table S6 explaining the key input parameters that we used in the MANDATE model analysis to estimate the number of deaths prevented per additional C section procedure performed when indicated for emergency delivery complications. We also added Figure 1 in the main text which shows how the MANDATE model fits into the overall analytic approach.

We added at Line 90:

See Table S6 for a detailed list of the baseline penetration and utilization rates of Caesarean section for each condition, as well as the corresponding efficacy parameters. We calculated the difference in caesarean delivery procedures performed (Table S6) and the difference in foetal and maternal deaths (Table S5), and then calculated a ratio of deaths prevented per additional caesarean delivery procedure.

RESPONSE: OK, now that I have the reference list, I was able to look up reference #16 and get a better understanding of the MANDATE model. While in most cases I do not think it is appropriate to present data outputs that have been generated externally, I do see the logic in doing that in this particular case. It helps tremendously that this MANDATE model has been peer reviewed. Your response is ok.

4. Why use life years in the denominator of the ICER if it is quite simple to use disability adjusted life years (DALYs)? The vast majority of cost-effectiveness analyses aimed at resource limited settings use DALYs in the denominator, which enables the reader to compare the value of interventions across therapeutic areas. It is not clear why the authors would not do the same.

Response: It is true that we would have preferred to use DALYs as our summary measure of health. This is the preferred outcome measure in CEAs in resource limited settings. However, we did not have data on the lifetime Years Lived with Disability (YLDs) a person in Kenya would experience if they did not die in childbirth. Moreover, the MANDATE model's scope only allowed us to estimate the number of deaths prevented by increased provision of Caesarean section. It does not estimate morbidity impacts. Finally, we believe the vast majority of the DALYs averted by preventing death in childbirth would be Years of Life Lost (YLLs) averted (i.e. survival gains). For these reasons, we used life-years instead of trying to calculate DALYs averted. Recognizing that the CEA value thresholds are based on DALYs, we did make a downward adjustment on the threshold used, since a life-year gained (if lived with some morbidity) is less valuable than a DALY averted.

RESPONSE: DALYS are the sum of YLD + YLL. You do not need to incorporate YLDs in your calculation to calculate DALYs, you can leave them at zero, in which case the YLLs you calculate using the MANDATE model are equivalent to DALYs. The life years gained in

your calculations are equivalent to a reduction in YLL, which are equivalent to a reduction in DALYs. In other words, you could replace the term “life years gained” in the text with the term “DALYs averted” and merely mention in the methods that the disability burden was not assessed and that the sole driver of DALYs averted were the expected gains in survival.

I think it would be better to present the results in terms of DALYs averted in order to allow for better comparisons to the cost-effectiveness of other maternal interventions in the literature. But this is more of a nice to have rather than a must have, I will vote to accept the paper even without this change. Perhaps the editor could weigh in?

5. The results do not add up. When you look at the national estimates, you have a total 5-year health system cost of \$1.2 million (page 12, line 37) and a benefit of 331,000 life years gained (page 13, line 44). That is \$3.62 per LY gained. However, in the cost-effectiveness section, the authors report a cost per life year gained of \$44 (page 14, line 7). How do I reconcile this difference?

Response: The \$1.2 million is the undiscounted 5-year program cost of the ESM-Ketamine program. To this must be added the undiscounted \$5.3 million extra delivery cost associated with additional Caesarean section deliveries over that 5-year period. (See Economic Outcomes panel of Table 3). When calculating the \$44 Cost per Life Year gained ICER shown in Table 4, we also discounted both costs and life years at 3% per year, as recommend in CEA guidelines.

\ RESPONSE: OK, back of the envelope I get \$20 when dividing \$6.5 million by 331,000, but since costs are incurred right away with little discounting and life years are discounted for much longer, the \$44 is plausible. OK.

Minor comments:

1. The authors define 20% of the Kenyan GDP to represent “good value”. This definition seems arbitrary. WHO-CHOICE defines an intervention to be highly cost-effective if the cost per DALY averted falls below one-time GDP. It is unclear where the 20% comes from. Response: We have added additional explanation regarding the choice of GDP-based cost-effectiveness threshold on page 10, line 132. In this, we acknowledge the WHO-CHOICE threshold of 1xGDP per capita per DALY averted, and explain why we use the more conservative health-opportunity loss approach, including a citation to a seminal paper on the topic. The source for the 25%*GDP per capita per DALY averted threshold we used is from the Beth Woods paper estimating health-opportunity-loss thresholds for low- and middle-income settings. Recently Ijeoma Edoka led an empirical analysis that found a threshold of 0.5*GDP per capita per DALY averted for South Africa – a much wealthier country than Kenya. We think this further supports the use of thresholds that are a fraction of GDP per capita in these settings. In any case, the intervention we evaluated was highly cost-effective even with these very conservative thresholds.

page 10, line 132:

Considerable debate persists about value thresholds for health interventions in settings like Kenya. The WHO has historically argued that interventions with an ICER less than per-capita GDP per DALY averted are cost-effective. More recently, researchers have advocated that much more stringent health-opportunity-loss based

	thresholds, are more appropriate when deciding to invest in new health technologies where health budgets are fixed [21]. \ RESPONSE: It still seems arbitrary and counter to the definition in WHO-CHOICE, but the references are very helpful. Ok. 2. Discussion, first paragraph, none of these numbers appear in the results section. It seems odd to discuss results that were not first presented in the results section. Response: The 28,700 enabled C-sections across 33 counties is reported in the Results section's first paragraph on page 12, as is 18% reduction in C Section gap. We added more explanation of the costs to the Results so that they are presented prior to the discussion. Page 12, line 177-182: The undiscounted cost of the five-year scale up of the ESM-Ketamine program is estimated to be about \$1.2 million (Table S3), or \$1700 per ESM-Ketamine provider trained. About half of this cost is the direct cost of training providers (Figure S1). The cost of the ESM-Ketamine program would be and \$41 per emergency caesarean delivery enabled. The total incremental cost, when including the additional cost of the caesarean procedure itself, is about \$224 per delivery. \ RESPONSE: OK
--	--